# The Linear Link: Deriving Age-Specific Death Rates from Life Expectancy

**Marius D. Pascariu** [1,2,*] , **Ugofilippo Basellini** [3,4] , **José Manuel Aburto** [2,3,5]
**and Vladimir Canudas-Romo** [6,*]

1    Biometric Risk Modelling Chapter, SCOR Global Life SE, 75795 Paris, France
2    Interdisciplinary Centre on Population Dynamics, University of Southern Denmark, 5000 Odense, Denmark
3    Max Planck Institute for Demographic Research (MPIDR), 18057 Rostock, Germany
4    Institut National D'études Démographiques (INED), 93300 Aubervilliers, France
5    Leverhulme Centre for Demographic Science, Department of Sociology and Nuffield College,
     University of Oxford, Oxford OX1 2BQ, UK
6    School of Demography, The Australian National University, Canberra 2600, Australia
*    Correspondence: mpascariu@pm.me (M.D.P.); vladimir.canudas-romo@anu.edu.au (V.C.-R.)

**Abstract:** The prediction of human longevity levels in the future by direct forecasting of life expectancy offers numerous advantages, compared to methods based on extrapolation of age-specific death rates. However, the reconstruction of accurate life tables starting from a given level of life expectancy at birth, or any other age, is not straightforward. Model life tables have been extensively used for estimating age patterns of mortality in poor-data countries. We propose a new model inspired by indirect estimation techniques applied in demography, which can be used to estimate full life tables at any point in time, based on a given value of life expectancy at birth. Our model relies on the existing high correlations between levels of life expectancy and death rates across ages. The methods presented in this paper are implemented in a publicly available R package.

**Keywords:** indirect estimation; life expectancy; forecasting; death rates; age patterns of mortality

## 1. Introduction

Understanding human mortality dynamics is of utmost importance in the context of rapid ageing process together with the increase in length of life experienced by most populations nowadays. The link between the pension systems sustainability and changes in life expectancy is more apparent than ever in light of the recent reforms that are taking place in Europe. In countries like Germany and Finland the level of retirement benefits are linked to life expectancy, in other countries like the U.K. and France the retirement age is set to increase from the current levels and implicitly the contribution period for pensions to be extended as people live longer (Stoeldraijer et al. 2013).

In predicting demographic processes, such as human mortality, methods involving extrapolation of mortality rates or probabilities are the most common approaches. Stochastic models, such as those proposed by Lee and Carter (1992) or Cairns et al. (2006) have gained significant popularity and have been extensively used in the last two decades. Ideas that focus only on life expectancy have given rise to a new approach. The models introduced by Torri and Vaupel (2012), Raftery et al. (2014) and Dan Pascariu et al. (2018) are partially inspired by the linear time trends observed in life expectancy at birth in many developed countries, particularly in the second half of the twentieth century

(Oeppen and Vaupel 2002; White 2002). These life expectancy models are very appealing because they offer the same, or higher, level of forecast accuracy in terms of life expectancy but with the advantage of being parsimonious, focusing on one variable rather than several. They rely on a measure that incorporates all the factors that influence longevity (lifestyle, access to healthcare, diet, economical status, etc.), namely life expectancy (Christensen et al. 2009). Furthermore, highly aggregated data by age provide valuable information that can be used to tackle the issue of mortality forecasting from a clearer perspective. The U.S. Census Bureau predicts the future mortality levels up to year 2100 based on projections of life expectancy at birth by sex and race, modelling an exponential decline of the gap to the observed upper asymptote of life expectancy. The period age-specific death rates are estimated in a subsequent step using these projections (United States Census Bureau 2014).

Transformation of life expectancy into mortality rates at every age can be accomplished by exploiting the regularities of age patterns of mortality. In actuarial science, the use of life tables and other models reflecting life contingencies is motivated by the need to determine insurance and pension risks, net premiums, and benefits. Basically, actuarial methods combine the life table with functions related to an assumed rate of interest (Dickson et al. 2013; Møller and Steffensen 2007). Based on the relevance of having a set of age-specific death rates, we propose a method to create such an array of values from one available life expectancy.

Our method extends the work initiated by the different systems of model life tables (Gabriel and Ronen 1958; United Nations 1955, 1967; Coale and Demeny 1966, Coale et al. 1983; Ledermann 1969; Sullivan 1972); Brass' relational model (Brass 1971; Brass et al. 1968) and the recent extensions of techniques for estimating age patterns of mortality by Murray et al. (2003) and Wilmoth et al. (2012). Our model is also related to the work of Mayhew and Smith (2013) that uses the trends in life expectancy to establish a robust statistical relation between changes in life expectancy and survivorship. A further, similar approach to the one developed here is that of Ševčíková et al. (2016) which incorporates a method based on the Lee-Carter model for converting projected life expectancies at birth to age-specific death rates in the UN's 2014 probabilistic population projections.

Relational models were developed for estimation purposes in poor-data contexts. These models rely on parameters that depict the relationships between various measures of age-specific and overall mortality. The parameters in a relational model are estimated from an initial analysis of historical mortality data and become fixed thereafter. Once those values have been estimated, the model simplifies to a few initial inputs like: reported child survival, records of population growth, responses to questions about fertility and mortality and in our case life expectancy at birth or at any other age. Our model recovers the entire age profile of mortality in a population based on the strong correlations between a single longevity measure, namely life expectancy, and age-specific death rates. Combining these correlations in the Lee-Carter (1992) methodology makes the proposed algorithm appealing to be used in forecasting practice. Furthermore, in recent years the accessibility of historical mortality data, such as the Human Mortality Database (HMD), means that the necessary information to estimate the parameters of the proposed model is readily available. Additionally, the linear relations between levels of life expectancy and age-specific death rates presented here, could be used for testing anomalies in information in actuarial practice.

The remainder of the article is organized as follows. First, in Section 2 a new model to derive age-specific death rates is introduced and a description of the data used in testing is provided. Section 3 shows computed results and illustrations of life expectancy decomposition into death rates in several populations, as well as comparisons of forecasts with other models. The discussion and conclusion are presented in Section 4.

## 2. Data and Methods

### 2.1. Data

The data source used in this article is the Human Mortality Database (University of California Berkeley, USA and Max Planck Institute for Demographic Research, Germany 2020), which contains historical mortality data for more than 50 homogeneous populations in 41 different countries and territories. The HMD constitutes a reliable data source because it includes high quality data that were subject to a uniform set of procedures, thus maintaining the cross-national comparability of the information.

In order to test and illustrate the performance of the method, we fit the model using the death rates computed using death counts and population exposed to the risk of death in the calendar year for the female populations of the England & Wales, France, Sweden and USA available in the HMD. The population selection was based on different degrees of model performance given by the mortality specificities of those populations, for example old age mortality in the HMD is often subject to diverse correction procedures and modelling depending on the country (Wilmoth et al. 2007).

The reconstructive power of the method for a point forecast of life expectancy is demonstrated using the 1980–2018 mortality data between age 0 and 100. Data at higher ages might be unreliable or too sparse for different populations, which would make it difficult to differentiate between data related problems or modelling issues. To compute the accuracy measures and the estimation errors, the 1965–90 data is applied to the same age range.

### 2.2. The Model

Given a predicted level of life expectancy the age pattern of mortality can be derived using a linear relation. The logarithm age-specific death rate at time $t$, denoted $m_{x,t}$, can be expressed as a linear function of the logarithm of life expectancy at a given age $\theta$, denoted $e_{\theta,t}$. Formally:

$$\log m_{x,t} = \beta_x \log e_{\theta,t} + \varepsilon_{x,t} \quad \text{for} \quad x \geq \theta, \tag{1}$$

where $x$ can take values between 0 and $\omega$, the highest attainable age, and $\beta_x$ can be regarded as an age-specific parameter. $\varepsilon_{x,t}$ denotes a set of normally distributed errors with mean zero and variance $\sigma^2$. For example, when $\theta$ equals zero, we estimate an entire mortality curve based on life expectancy at birth using this equation; when $\theta > 0$, we estimate the mortality curve starting from age $\theta$.

The method presented here combines the linear relations found when comparing life expectancies and age-specific death rates on a log-log scale. Figures 1 and 2 show those relations, although the slopes and intercepts vary, in all cases there is a significant linear concordance between the level of overall mortality, as depicted by life expectancy, and the individual age-specific death rates especially between age 30 and 90 as displayed by the Pearson correlation coefficient. These relations have been key in much of the work on model life tables (Gabriel and Ronen 1958; United Nations 1955, 1967; Coale and Demeny 1966, Coale et al. 1983; Ledermann 1969). Inspired by the two-dimensional system (age and time) of the Log-quadratic model Wilmoth et al. (2012) and the strong linear trends in Figures 1 and 2, we derive the age pattern of mortality based on a given value of life expectancy, e.g., forecasted life expectancy value, and a matrix of age-specific death rates from the past.

This model can be seen as a method that links the life expectancy at age $\theta$ at any point in time to a mortality curve estimated from the death rates $m_x$'s that return a life expectancy level of $e_\theta$. Therefore we will refer to it as the linear-link (LL) model. To gain precision in the fitting of the death rates the LL model can be extended by including additional parameters:

$$\log m_{x,t} = \beta_x \log e_{\theta,t} + \nu_x k + \varepsilon_{x,t} \quad \text{for} \quad x \geq \theta,$$

$$\sum_{x=\theta}^{\omega} \nu_x = 1, \quad \text{and} \quad \nu_x \geq 0, \tag{2}$$

where $\nu_x$ is the speed of mortality improvement over time at age $x$, $k$ is an estimated correction factor independent of time and $\varepsilon_{x,t}$ are independent and identically distributed random variables normally distributed with mean zero and variance $\sigma^2$. Different than the Log-quadratic model that has a fixed set of parameters for any input value, here the parameters $\beta_x$, $\nu_x$ and $k$ can be calculated for each set of age-specific death rates and future life expectancy. Thus, it can be seen as an extension of the log-quadratic model for countries that have good quality data, where an entire life table is completed from one target value of life expectancy.

In addition, the LL model is closely related to the LC model. Indeed, if one sets the parameters $\beta_x \log e_{\theta,t} = \alpha_x$, $\nu_x = \beta_x$, and $k = k_t$, we obtain the LC model. Interpretation of the parameters are then similar with $\beta_x \log e_{\theta,t}$ a standard age profile, $\nu_x$ the age-specific improvements in mortality, and $k$ the amount of average mortality improvement. Despite their similarities, there are two important differences between the two models. First, while the shape of the mortality pattern $\alpha_x$ is constant in the LC model, the first term of the LL changes with the level of life expectancy considered; as such, there exists a range of different baseline mortality curves of the LL model depending on the particular level of $e_\theta$. Second, the $k$ parameter is not modelled as a function of time, instead the parameter is used as an optimization variable affecting the shape of the age pattern of mortality to achieve the desired target life expectancy $e_\theta$. Thus, the $k$ parameter enhances the flexibility of the method and the accuracy of the results.

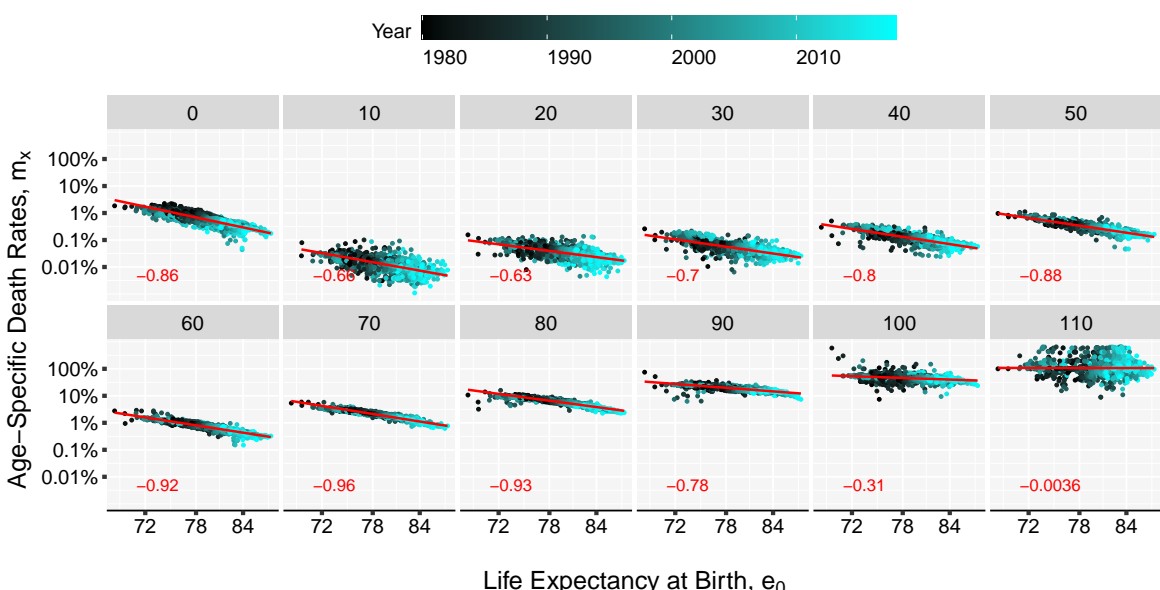

**Figure 1.** Linear relation between life expectancy at birth and death-rates on a log-log scale, by age displayed together with the Pearson correlation coefficient, in the bottom left end of the panels. Each panel contains data for a specific age. The axis are labelled in normal scale for better interpretability. Based on HMD mortality data starting from 1980 for 41 countries and territories.

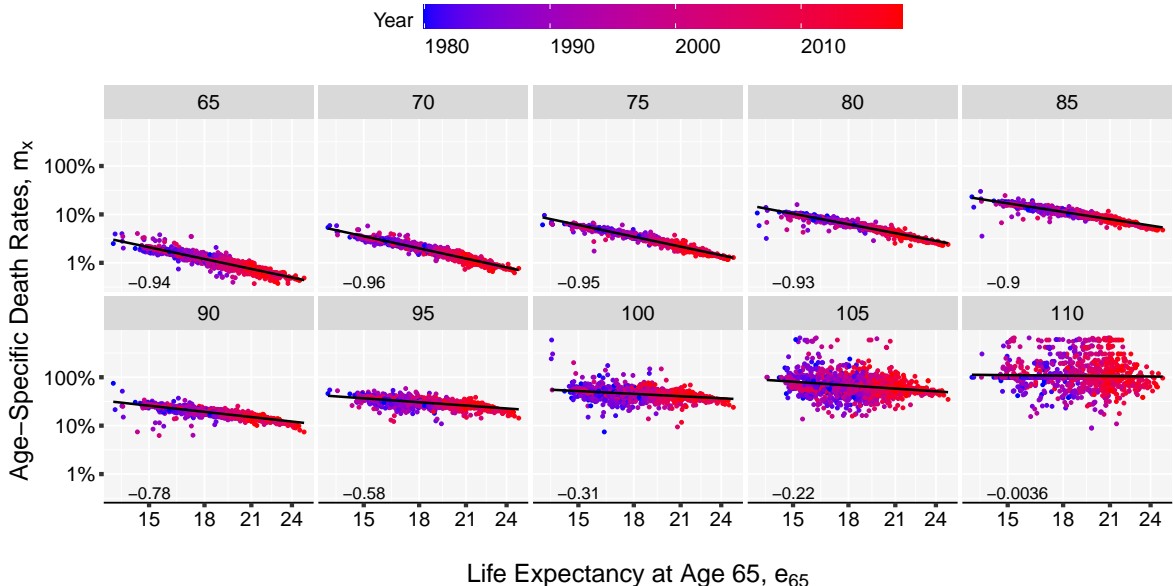

**Figure 2.** Linear relation between life expectancy at age 65 and death-rates on a log-log scale, by age displayed together with the Pearson correlation coefficient, in the bottom left end of the panels. Each panel contains data for a specific age. The axis are labelled in normal scale for better interpretability. Based on HMD mortality data starting from 1980 for 41 countries and territories.

### 2.3. Algorithm

Let $t$ be an observed unit of time in the interval $\{1, ..., T\}$ and $\tau$ be a an unobserved point in time $T + n$ e.g., a date in the future. The objective is to convert a value of life expectancy, $e^*_{\theta,\tau}$, into a schedule of age-specific death rates $m^*_{x,\tau}$. The level of life expectancy can be a predicted value given by certain extrapolation method or the target values resulted following a subjective judgement. Input data will be a collection of observed death rates $m_{x,t}$ and a level of life expectancy $e^*_{\theta,\tau}$. The steps involved in the algorithm to obtain the desired death rates are the following:

1. Using the Kannisto mortality model (see Appendix A) extend $m_{x,t}$ to higher age groups up to age $\omega$ for all times $t$. The highest attainable age, $\omega$, can be set for example to 120.
2. Estimate the slope of the linear relation between life expectancy and the death-rates, $\beta_x$, over the observation time $t$. This is done by using the method of the least squares approach, by minimizing the sum of squared residuals:

$$\sum_x [\log m_{x,t} - \beta_x \log e_{\theta,t}]^2 = \sum_x [\varepsilon_{x,t}]^2. \tag{3}$$

Alternatively, the parameters of the model can be estimated by assuming that deaths follow a Poisson distribution (Brillinger 1986; Brouhns et al. 2002), $D_x \sim Poisson(E^c_x \cdot m_{x,t})$, with $m_{x,t} = \exp(\beta_x \log e_\theta + v_x k)$. In order to use this approach death counts ($D_{x,t}$) and central exposure data ($E^c_{x,t}$) are needed. Sensitivity analysis shows that the difference between the two fitting procedure return minor discrepancies (see Appendix B in the Appendix for more details).

3. Estimate the parameter $v_x$ by computing the singular value decomposition (SVD) of the matrix of regression residuals, **R**, obtained in the previous step,

$$SVD\,[\mathbf{R}] = \mathbf{DPQ}^T = d_1 p_1 q_1^T + \ldots, \tag{4}$$

where

$$\mathbf{R} = \begin{bmatrix} \varepsilon_{0,1} & \varepsilon_{0,2} & \cdots & \varepsilon_{0,T} \\ \varepsilon_{1,1} & \varepsilon_{1,2} & \cdots & \varepsilon_{1,T} \\ \vdots & \vdots & \ddots & \vdots \\ \varepsilon_{\omega,1} & \varepsilon_{\omega,2} & \cdots & \varepsilon_{\omega,T} \end{bmatrix},$$

$\mathbf{P} = [p_1, p_2, \ldots]$ and $\mathbf{Q} = [q_1, q_2, \ldots]$ are matrices of left and right singular vectors, and $\mathbf{D}$ is a diagonal matrix with singular values along the diagonal. The fist term of the $SVD$, $d_1 p_1 q_1^T$, is used for obtaining the estimates of $v_x$. Parameter $v_x$ can be interpreted as the rate of mortality improvement over age.

4. Smooth the $\beta_x$ and $v_x$ parameters using splines. This step is important to obtain graduated mortality curves and avoid projecting age-specific noise in the jump-off life table. However, if the graduation is not of interest or if the input data-set is large enough, this step can be skipped.

5. Compute the initial mortality rates[1] by $m_{x,\tau}^* = \exp\{\beta_x \log e_{\theta,\tau}^* + v_x k\}$, where $k = 0$.

6. Optimize the mortality curve given in the previous step by finding the value of $k$ where the difference between target life expectancy $e_{\theta,\tau}^*$ and an estimated life expectancy $e_{\theta,\tau}$ is below a tolerance level, for example 0.001, where $e_{\theta,\tau}$ represents the level of life expectancy at birth computed based on the mortality rates obtained in step (5). Usually $k$ will be in the range of $(-150, +150)$ depending on the length of the forecast window.

The estimated $\beta$ parameters for the female populations in England & Wales, France, Sweden and USA, exhibit minor differences between the countries, and capture well the important stages of human mortality: the decreasing infant mortality, the accidental hump, the adult mortality characterized by an exponential increase with age and, finally, a mortality plateau above the age of 100 years. As shown in Figure 3, the $v_x$ pattern differs from population to population. In the case of Sweden, a larger variance is observed over ages due to a smaller population size and more significant changes at younger ages in the period analysed.

---

[1] The change in age-specific death rates can be assumed to be constant over time, in which case the fitted $v_x$ is used in computing $m_x$. Or, a shift in the speed of improvement can be imposed by "rotating" the $v_x$ coefficients. For more details see Section in the Appendix C.

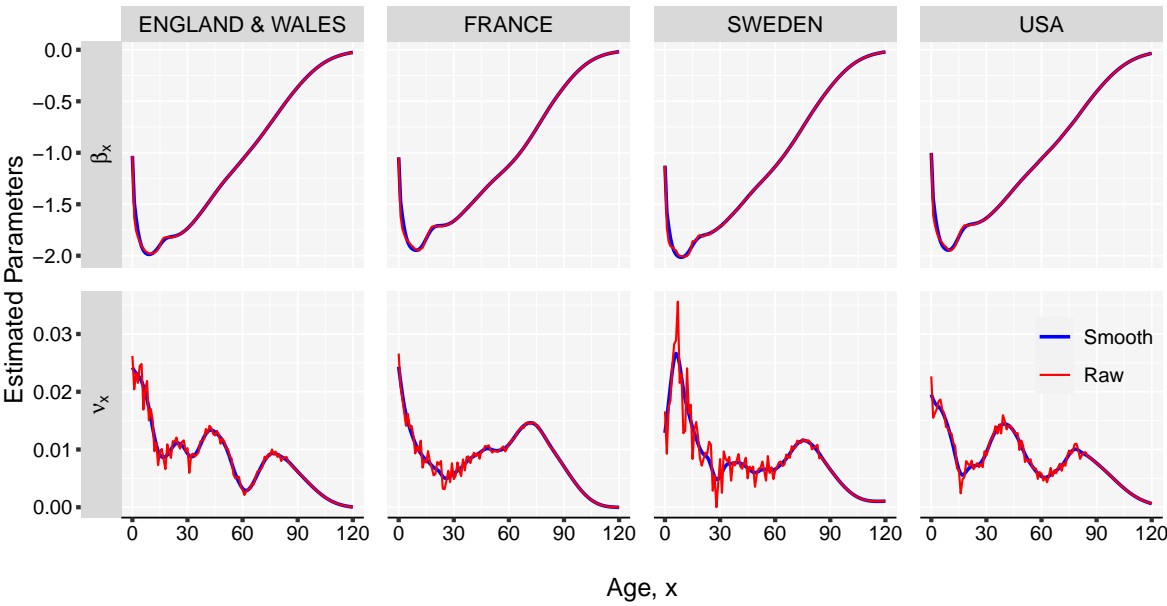

**Figure 3.** Estimated parameters of the Linear-link model, using HMD data from 1980 to 2018 and life expectancy at birth ($\theta = 0$).

## 3. Results and Illustration

We perform back-testing against the observed mortality for the female populations living in England & Wales, France, Sweden and USA. We take the period of 1965–1990 as reference and use the death-rates and life expectancies at birth in this time interval to fit our model. Based on single values of life expectancy at birth observed in the subsequent years we derive complete mortality curves. For example, the estimation of the age-specific death rates in 2018 is demonstrated in Figure 4. The reconstructed mortality curves are in general smoother than the observed data; this is more evident in the case of Sweden, where the population is smaller compared with the other three countries.

Figure 5 shows that the average relative error of the estimated log-death rates, compared to the actual rates between 1991 and 2018, is between 0.9% and 4.3%. It can be also noted that the longer the prediction interval, the larger the errors. In the case of female populations living in England & Wales, France, Sweden and USA, the largest errors occurred in 2016; nonetheless these are smaller than 4.3% of the actual log-death rate. This value is an average over the entire comparable age range (0–100). The largest impact on the overall accuracy occurs at advanced ages, where the level of uncertainty is higher. Figure 6 offers a view of the error distribution by age and time. However, the life expectancy at birth computed based on the estimated death rates matches exactly the actual life expectancy in the respective year and country.

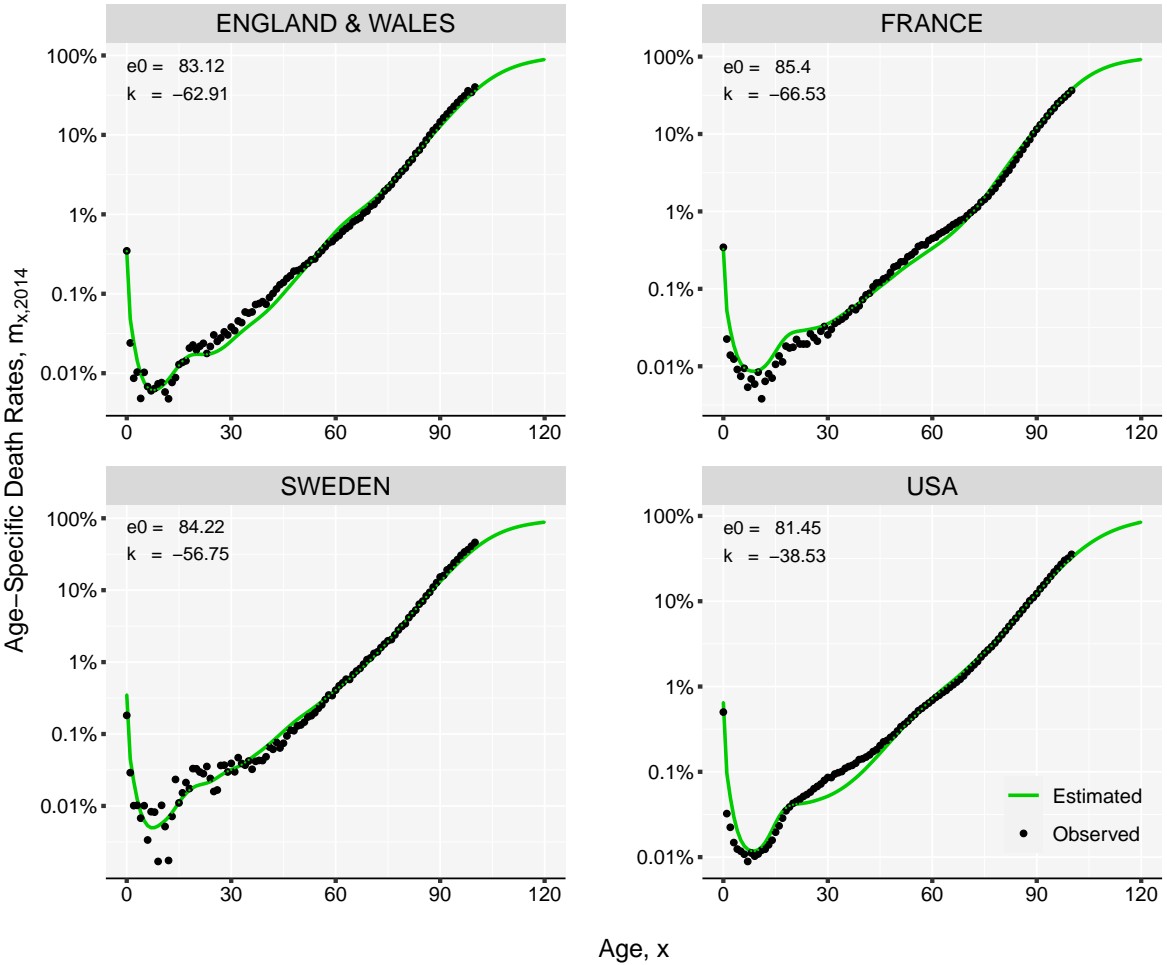

**Figure 4.** Observed and estimated death rates for female populations in 2018. Computed based on mortality data in the period 1965–1990.

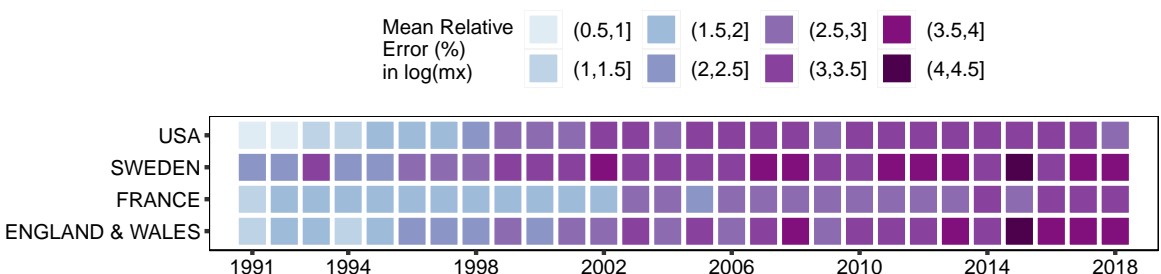

**Figure 5.** Mean absolute errors (%) of the estimated log-death rates against the actual log-death rates between 1991 and 2018. Computed based on female mortality data in the period 1965–1990.

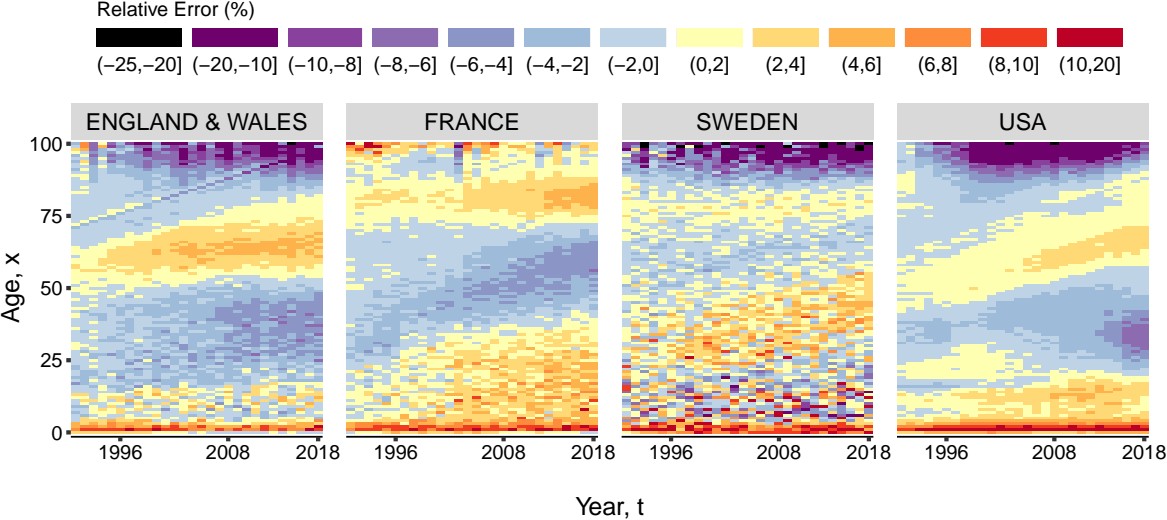

**Figure 6.** Relative errors (%) of the estimated log-death rates against the actual log-death rates between 1991 and 2018. Computed as (observed-estimated)/ observed log death rates. Based on female mortality data in the period 1965–1990.

In order to test the conversion reliability of a forecast value of life expectancy, we compare the results generated by the LL model against the predicted mortality from the Lee-Carter model (1992).

The LC model is fitted over the 0–95 age-range using the historical data from 1980 to 2018, and used to forecast death rates 22 years in the future, until 2040. The estimated matrix of predicted death rates between age 0 and age 95 is extended up to age 120 using the Kannisto model (see Equation (A1) in the Appendix A). If multiple projections are simulated for the same forecast point, the LC would produce a range of outcomes that can be translated into life expectancies using standard life table calculations. Any predicted life expectancy given by LC is used as an input value in the LL model to derive the mortality curve, thus obtaining two comparable curves. For every simulated trajectory, the LL method can produce a mortality curve, generating the uncertainty around the median prediction. Figure 7 shows that the reconstruction method employed by the LL model gives an almost coincident mortality curve when compared with the LC curve for female populations in 2040. The 99% prediction intervals are computed based on ten thousand Monte-Carlo simulations. Besides the LL model showing a more smooth age-pattern when compared with the LC results, it can perfectly estimate the predetermined life expectancy, and it exploits the linear relationship between mortality and life expectancy producing mortality profiles that are less distorted around the age dimensions when forecasted far into the future.

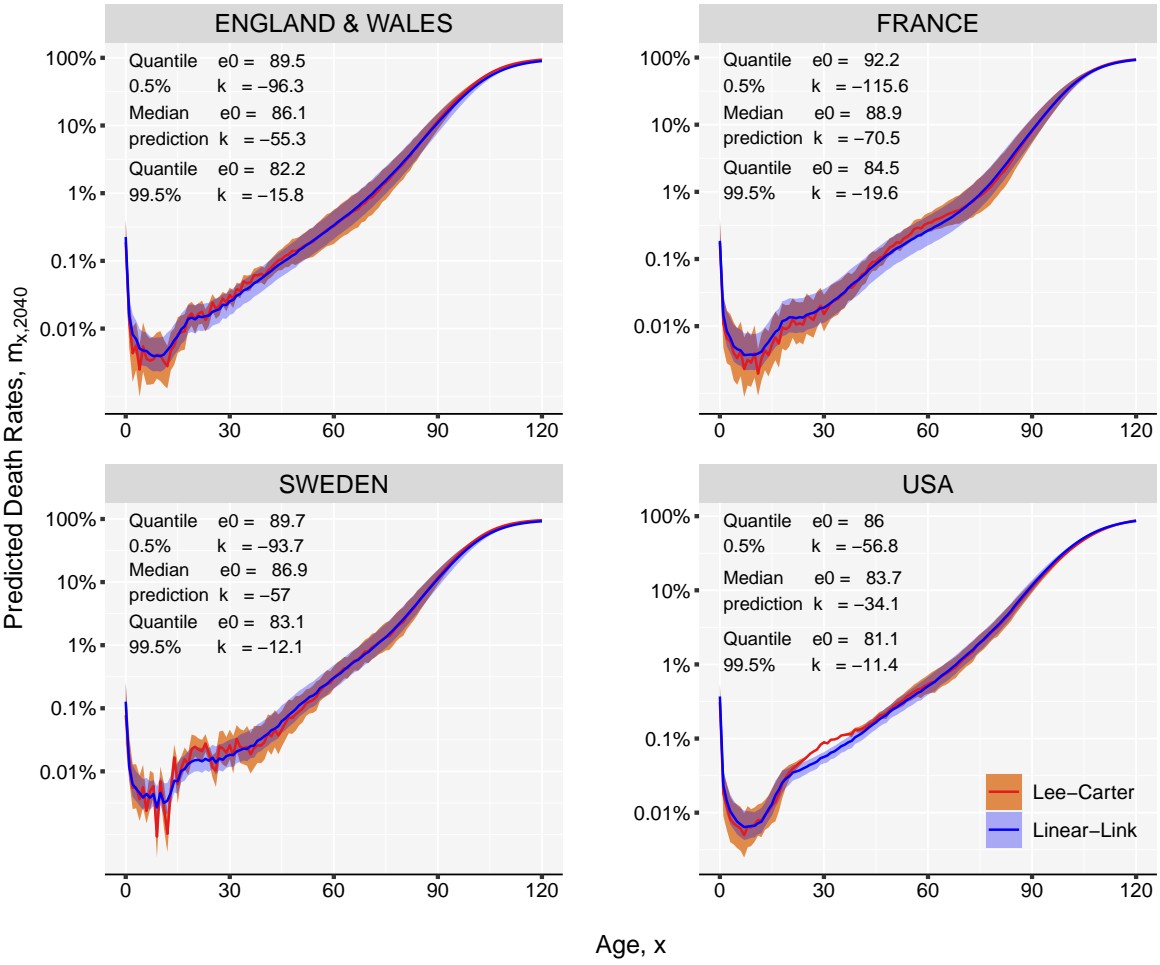

**Figure 7.** Comparison of the mortality curves predicted by Lee-Carter and Linear-Link models in 2040 from female populations. The models are fitted on the 1980–2018 historical period.

## 4. Discussion

We have introduced a simple method, the Linear-link model, to derive the entire schedule of age-specific death rates, based on a single value of life expectancy and prior knowledge of human mortality patterns. The model is based on the observed linearity between age-specific death rates, $m_x$, and life expectancy at a certain age, $e_\theta$. The model can be regarded as a decomposition approach of the human mortality curve between the general age pattern, $\beta_x$, and an age-specific speed of improvement, $\nu_x$. The method is inspired by: (1) the Log-quadratic model (Wilmoth et al. 2012) in the sense of using a leading indicator in determining the age pattern of mortality; (2) the model introduced by Ševčíková et al. (2016) by adopting an inverse approach to death rates estimation starting from life expectancy; (3) the Lee-Carter model (1992) using the same interpretation of mortality improvement over time and age; and finally (4) the Li et al. (2013) method to model the rotation of age patterns of mortality decline for long-term projections.

The method can be useful in three different situations: future target life expectancy, life tables for countries with deficient data and historical life table construction. The former is the one explored in the present manuscript, while the latter two are only briefly discussed since their development goes beyond the scope of the paper of presenting the LL model. Future work is envisioned to apply and compare the proposed Linear-link with other approaches in contexts of lower data quality or availability.

First the model can be used in forecasting practice when the level of life expectancy is forecast first. We showed that this model can accurately reconstruct a Lee-Carter forecast starting from a single value of life expectancy at birth. This is important, because the Linear-link model offers the possibility of taking advantage of the more regular pattern of the life expectancy evolution. It is much easier and parsimonious, from a technical perspective to forecast one time series of expectation of life than to extrapolate 100 or 110 series of death probabilities corresponding to each age group. In the same manner adult mortality can be estimated based on a value of life expectancy at an advanced age, say age 65. In Figure 2, we showed that the linearity between death rates at advance ages and life expectancy at age 65 on a log scale is maintained. A greater variation is observed only at advanced ages, above 100, where data is sparse in general.

Second, the method can be used to build model life tables and estimate the current age patterns of mortality in poor-data countries or regions, like Sub-Saharan Africa or Central Asia. National and international agencies usually make use of indirect estimation methods heavily based on expert judgment to determine demographic indicators and compensate for the lack of data. Relational models where one indicator (e.g., life expectancy of the population) is based on another (e.g., survival rates in a specific age range) are common. The LL method intends to provide a new solution within this framework i.e. the transition from a life expectancy measure to an age specific mortality pattern. As in the case of relational models the method alone cannot represent a solution for all data-deficient regions however, in the hands of a qualified country specialist capable of making an educated guess on life expectancy and borrow information from populations with identical characteristics and good quality data the advantages of the Linear-Link method are obvious. The parameters of the model can be estimated in this case based on a collection of historical life tables from several regions or populations, possibly with higher data quality. Once the parameters have been estimated, and implicitly the model life table, they remain fixed. The relevant mortality curve is simply calibrated in accordance with a single value of life expectancy at birth or any other age instead of child mortality like in the case of Wilmoth et al. (2012). In our analysis we show examples using high quality data from developed countries in order to demonstrate the efficiency of the model, and to be able to assess the accuracy of the mortality curve reconstruction. However, the estimation procedure and the steps of the algorithm are the same for this case too.

Third, the LL model can be a useful tool in a variety of research contexts of historical demography like backward projections and estimation of mortality levels in historical populations. Due to the existence of scarce non-standardized population data in the past and population census only for the more recent times, the very possibility of projecting mortality backward is of theoretical interest (Ediev 2011).

There are two limitations of our proposed approach that should be mentioned: (i) its dependency on age-specific mortality information, which is needed to estimate the model's parameters, and (ii) the accuracy of such information. The former one is particularly relevant for countries with lower data quality and availability, and it can be overcome by borrowing information from neighboring regions, as in the spirit of model life tables. The second limitation is more generally shared by any methodology that aims at modeling and forecasting mortality age-patterns.

According to our analysis, the optimal number of years to be used in the fitting of the model is between 30 and 35 years. If a longer time interval was used, the parameter estimates would lose their relevance. For example, the present rate of improvement in the death rates is different from that experienced 50 years ago, because of fundamental changes in society and scientific advances during this period (Bengtsson 2006; Rau et al. 2008). In the same manner over a longer period of time the linearity between life expectancy and death rates might be challenged, however this should be investigated from case to case.

The speed of improvement in age-specific death rates over ages changes over time. For example, in recent decades, a faster pace of improvement was observed at ages 65 and above (Shkolnikov et al. 2011; Vaupel 1997). We address the possibility of experiencing accelerating or decelerating speeds of mortality improvements over different age ranges by assigning different weights to the estimated $\nu_x$ curve when the

life expectancy at birth continues to advance over age 75. The effect of this method can be best observed in Figure 7 in the case of France. Under the implicit assumption of constant mortality improvements, the LC forecast generates a second mortality hump around age 50. The estimated mortality curve given by the LL model has a less pronounced effect due to the rotated $\nu_x$ parameter. See a detailed description of the method in Appendix C.

The evolution of human mortality is a complex process that is driven by a large number of factors and can not be explained by a single statistical model. The Linear-link method offers an alternative approach to deriving the unknown levels of mortality in the future. In contrast with methods like the Lee-Carter model that extrapolate age-specific rates or probabilities directly the method presented here recognizes life expectancy as the main driver of mortality at any given age and employs an indirect estimation algorithm. These methods can complement each other and help us understand better the future longevity experienced by populations.

## 5. Reproducible Research

The presented model and algorithm is implemented using the R programming language (R Core Team 2019) and can be downloaded and installed in form of an R software package from authors' GitHub repository. The results and figures for the four countries presented in this article can be reproduced using the code and data saved in the R package.

**Author Contributions:** Conceptualization, M.D.P.; methodology, M.D.P. and V.C.-R.; software, M.D.P., U.B. and J.M.A.; validation, V.C.-R., J.M.A. and U.B.; formal analysis, M.D.P.; investigation, M.D.P.; writing–original draft preparation, M.D.P.; writing–review and editing, V.C.-R., J.M.A. and U.B.; visualization, M.D.P. and U.B.; supervision, V.C.-R.; funding acquisition, M.D.P. All authors have read and agreed to the published version of the manuscript.

**Funding:** This work was conducted within the "Modelling and Forecasting Age-Specific Death at Older Age", Project [No. 95-103-31186], under the management of the University of Southern Denmark, Institute of Public Health with the generous financial support of the SCOR Corporate Foundation for Science. The authors thank the funding institution. J.M.A was supported by the British Academy's Newton International Fellowship.

**Conflicts of Interest:** The authors declare no conflict of interest. The funders had no role in the design of the study; in the collection, analyses, or interpretation of data; in the writing of the manuscript, or in the decision to publish the results.

## Appendix A. The Kannisto Model

Normally, mortality data is available in tables that contain detailed information up to age 85, 100 or 110, with last age group being open. In order to extend the mortality rates up to age 120, the Kannisto method (Thatcher et al. 1998) for old-age mortality with an asymptote equal to one can be employed:

$$m_x = \frac{\alpha e^{\beta x}}{1 + \alpha e^{\beta x}}, \tag{A1}$$

which can also be written as a linear function of age

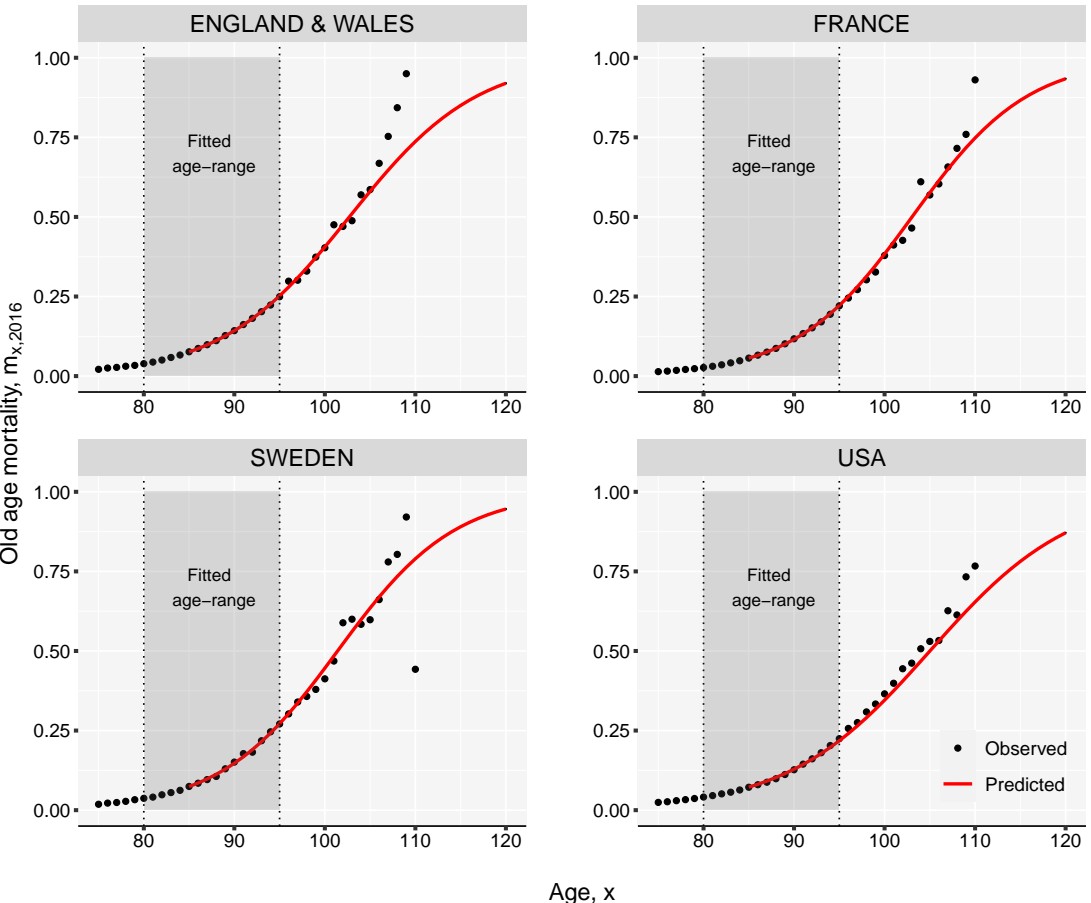

**Figure A1.** Extension of female mortality rates using the Kannisto model in 2016.

$$logit(m_x) = ln(\alpha) + \beta\chi + \epsilon_\chi, \tag{A2}$$

where $\epsilon$ is a normally distributed variable with mean zero, $\chi = x - 80$, and parameters $\alpha$ and $\beta$ are positive real numbers. The model is usually fitted between age 80 and 95.

Assuming that $D_x \backsim Poisson(E_x^c \cdot m_x(\alpha, \beta))$ the parameters $\alpha$ and $\beta$ can be derived by maximizing the log-likelihood function:

$$logL(\alpha, \beta) = \sum_{x=80}^{95} \left\{ D_x log[m_x(\alpha, \beta)] - E_x^c m_x(\alpha, \beta) \right\} + constant, \tag{A3}$$

where $D_x$ denotes the number of deaths that occurred at age $x$, $E_x^c$ represents the population exposed to risk at the same age, and $m_x$ is the age-specific death rate.

The Kannisto model is not only useful to obtain values for oldest-old mortality but also to smooth the rates computed on smaller sample sizes. The case of Sweden presented in Figure A1 can be relevant here, where in 2016 the number of females aged 100 and above was less than 1700. A small sample size can create difficulties in obtaining reliable mortality estimates based only on empirical observations. Outliers are expected to show up from year to year.

## Appendix B. Maximum Likelihood Estimation

Assuming that deaths are Poisson distributed, the LL model can be fitted by maximising the log-likelihood given by

$$logL(\beta, \nu, k) = \sum_{x,t} \left\{ D_{x,t}(\beta_x \log e_{\theta,t} + \nu_x k) - E_{x,t}^c \exp(\beta_x \log e_{\theta,t} + \nu_x k) \right\} + C, \tag{A4}$$

where $C$ is a constant. The parameters are estimated following an updating scheme proposed by Brouhns et al. (2002) based on the Newton-Raphson algorithm. The updating procedure, with initial values $\hat{\beta}_x(0) = 0$, $\hat{\nu}_x(0) = 1$, and $\hat{k}(0) = 0$, is as follows:

$$\hat{\beta}_{x,w+1} = \hat{\beta}_{x,w} + \frac{\sum_t (D_{x,t} - \hat{D}_{x,t,w})(\log e_{\theta,t})}{\sum_t \hat{D}_{x,t,w}(\log e_{\theta,t})^2},$$

$$\hat{\nu}_{x,w+1} = \hat{\nu}_{x,w},$$

$$\hat{k}_{w+1} = \hat{k}_w,$$

$$\hat{k}_{w+2} = \hat{k}_{w+1} + \frac{\sum_x (D_{x,t} - \hat{D}_{x,t,w+1})(\log e_{\theta,t})(\hat{\nu}_{x,w+1})}{\sum_x \hat{D}_{x,t,w}(\log e_{\theta,t})^2(\hat{\nu}_{x,w+1})^2},$$

$$\hat{\beta}_{x,w+2} = \hat{\beta}_{x,w+1},$$

$$\hat{\nu}_{x,w+2} = \hat{\nu}_{x,w+1},$$

$$\hat{\nu}_{x,w+3} = \hat{\nu}_{x,w+2} + \frac{\sum_t (D_{x,t} - \hat{D}_{x,t,w+2})(\log e_{\theta,t})(\hat{k}_{w+2})}{\sum_t \hat{D}_{x,t,w+2}(\log e_{\theta,t})^2(\hat{k}_{w+2})^2},$$

$$\hat{\beta}_{x,w+3} = \hat{\beta}_{x,w+2},$$

$$\hat{k}_{w+3} = \hat{k}_{w+2},$$

where $\hat{D}_{x,t,w} = E_{x,t}^c \exp(\hat{\beta}_{x,w} \log e_{\theta,t} + \hat{\nu}_{x,w} \hat{k}_w)$, is the estimated number of deaths after iteration $w$.

The maximum likelihood estimation (MLE) has several advantages over least squares (OLS) and SVD methods or even weighted least squares (WLS) used in Wilmoth et al. (2007). Several reasons have been given in the literature (Alho 2000; Brouhns et al. 2002). One example would be the increasing confidence intervals by age. This is because in the OLS estimation via SVD the errors are assumed to be homoskedastic and normally distributed, which is quite a heavy assumption. The logarithm of the observed force of mortality is much more variable at older ages than at younger ages because of the much smaller absolute number of deaths at older ages. Therefore, since the number of deaths is a counting variable, the Poisson assumption seems more reasonable (Brillinger 1986).

However, in order to use this approach we need death counts $D_{x,t}$ and exposures $E_{x,t}^c$, which are not always available. This being the reason why the model described by Equation (1) is chosen in the article. Our methodology is targeting populations with deficient data as well as populations whose estimates of mortality rates are provided without disaggregation by deaths and exposures. Although conceptually a Poisson setting would be better, the SVD approach is a pragmatic decision for practical reasons. As shown in figure A2 the difference between the two estimation methods for the case of England & Wales females is very small but the data requirements is higher in the case of MLE.

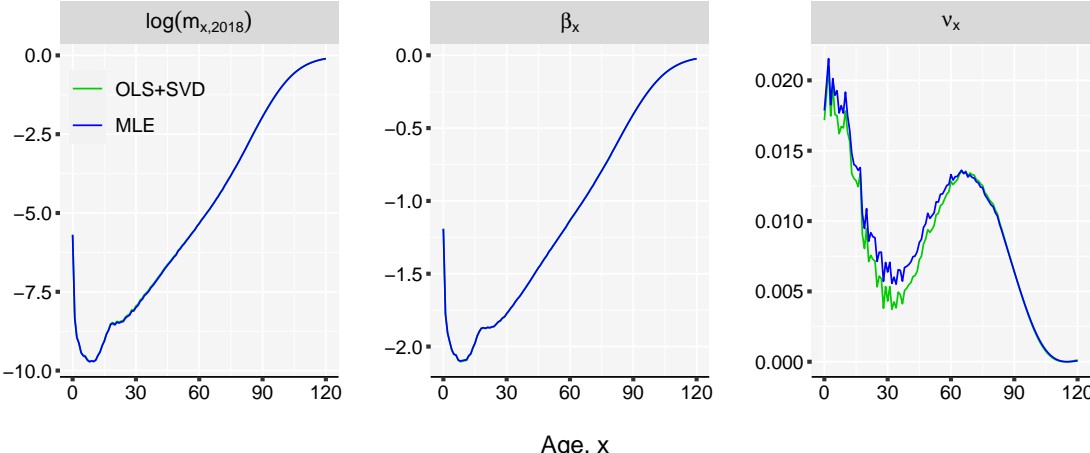

**Figure A2.** Comparison of the fitted mortality curves and parameter estimates of the Linear-Link model using the OLS+SVD and MLE fitting procedures. England & Wales female data for 1980–2018 period is used.

## Appendix C. Rotation of Mortality Improvements

One of the main limitations of the LC model (1992) is the central assumption of constant rates of mortality declines at different ages, resulting from the time-invariant $b_x$ coefficient of age-specific mortality improvements (Bongaarts 2005). The assumption has been violated in several low-mortality countries in recent decades, because rates of mortality improvements have tended to decline over time at younger ages, and they have risen at older ages (Kannisto et al. 1994; Vaupel et al. 1998; Wilmoth and Horiuchi 1999).

It is important to take into consideration the changing age pattern of mortality improvements to produce more accurate mortality forecasts, and projection methodologies that ignore such rotation will lead to errors, particularly in the projected age patterns of future death rates (Li et al. 2013). Li et al. proposed an extension of the LC method to incorporate the rotation of the age patterns of mortality decline for long-term projections.

Here, we propose a modification of the original Li et al. (2013) methodology that ensures the rotation of the rate of mortality improvement over age in the LL model, $v_x$. The methodology is composed of two different steps.

First, we derive an ultimate schedule of mortality improvements, $v_x^u$, from the estimated coefficient $v_x$. In particular, the ultimate rates of improvement between ages 0 and 65 are set equal to the average improvement at adolescent and adult ages (15–65); from age 65 onwards, improvements decrease following a logistic shape, and they converge to zero at age 130.

Second, we smooth the transition from $v_x$ to $v_x^u$ using the weight function proposed by Li et al. (2013). The transition, and therefore the degree of rotation of $v_x$, is dependent on $e_0^*(\tau)$, the predicted value of life expectancy at birth (an input in our LL model). Formally, the weight function $w_s$ can be expressed as:

$$w_s(\tau) = \left\{ \frac{1}{2} \left[ 1 + \sin \left[ \frac{\pi}{2} \left( 2w(\tau) - 1 \right) \right] \right] \right\}^p \quad \text{with} \quad w(t) = \frac{e_0^*(\tau) - 80}{e_0^u - 80}. \tag{A5}$$

Because $v_x$ parameter is scaled in order to take values between 0 and 1 the ultimate pattern of mortality improvement by age will be the same for all countries. If the scaling process is ignored, the same pattern is obtained in all cases with a different level of $v_x$ between age 0 and 65. The estimated death rates would be the same in both case because of the adjustment provided by $k$ parameter.

The power of the smooth-weight function, $p$, regulates the speed of the rotation. It varies between 0 and 1, and lower values correspond to faster rotations for levels of $e_0^*(\tau)$ closer to 80. The level of life expectancy at which the rotation finishes, $e_0^u$, is also arbitrary; here, we follow the recommendations of Li et al. (2013) and set the intermediate value of 0.5 for $p$ and the age 102 for $e_0^u$.

The rotated coefficient of mortality improvement over age, denoted $N_x^r(\tau)$, can thus be written as:

$$N_x^r(\tau) = \begin{cases} \nu_x \,, & e_0^*(\tau) < 80 \,, \\ [1 - w_s(\tau)] \, \nu_x + w_s(\tau) \nu_x^u \,, & 80 \leq e_0^*(\tau) < e_0^u \,, \\ \nu_x^u \,, & e_0^*(\tau) \geq e_0^u \,. \end{cases} \tag{A6}$$

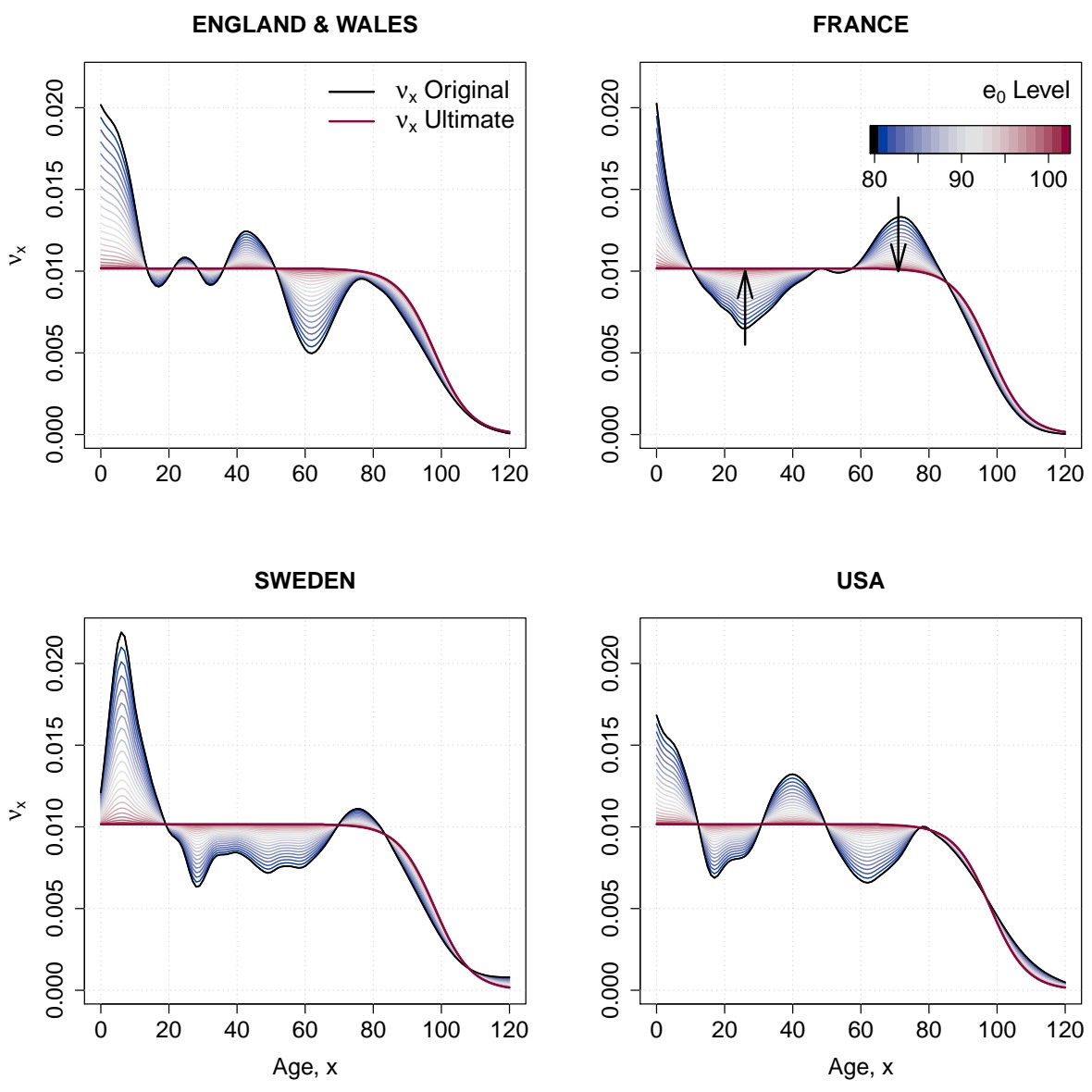

**Figure A3.** Assumption of the change in $\nu_x$ pattern following the increase in life expectancy at birth from 75 to 102 years.

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
