# Peer review of "The Linear Link: Deriving Age-Specific Death Rates from Life Expectancy"

_risks, doi:10.3390/risks8040109_

Round 1

Reviewer 1 Report

See attached

Author Response

See file attached.

Reviewer 2 Report

The paper submitted by Pascariu et al. can be considered within the scientific area of probabilistic models used to estimate mortality rates in human populations.

The authors presented a new mathematical method, the Linear-link model, based on the observation that a linear relation between age-specific death rates and  life expectancy exists. Starting from this consideration, the algorithm is developed and demonstrates that full time tables at any time can be forecasted if only one parameter is calculated, i.e.  life expectancy at birth.

The paper is well written, the authors reviewed many similar methods previously published to which they compared their new analysis and they described the mathematical algorithm extensively and clearly. They tested the method on real data and they obtain a good fit between observed and expected results  strongly demonstrating the assumptions the authors started from to develop the model.

The idea of the paper is very interesting and the results the Authors have achieved provide a new relational model that seems to be innovative, simpler and more parsimonious respect to the previous proposed algorithms

However, in my opinion, some minor revisions has to be made in order to improve the manuscript:

Rows 38

The bibliographic list of papers cited is not completed. I suggest to insert the correct bibliographic reference and to control carefully the bibliography inserted all over the paper. 

Figure 4

The colours chosen to represent the estimated and observed death rates for female populations in 2016 are too similar and they do not contrast sufficiently to allow a correct view of the difference between the estimates and the real data. I suggest to change those colours similar to the ones present in figure 3.

Discussion

Most discussion is concentrated to describe applications of the model to situations that do not belong to the goal of the paper, as stated by the Authors. I suggest to reduce or eliminate that part and to concentrate the discussion only to the results presented in paper (see rows 218-245).

Author Response

See file attached.

Reviewer 3 Report

My review is based on a general comment and further particular comments.

As a general comment, this paper proposes a new relational model. This model is inspired by indirect estimation techniques applied in demography, which can be used to estimate full life tables at any point in time, based on a given value of life expectancy at birth or life expectancy at other age. This new model is useful for age patterns of mortality in poor-data countries; nevertheless, authors fit the model using the death rates computed using death counts and population exposed to the risk of death in the calendar year for the female populations of the England &Wales, France, Sweden and USA available in the HMD.

In my opinion, the main drawbacks are that the new model should test linearity in the relation between age-specific death mortality and life expectancy, and prediction should be using confidence intervals.

The authors should also state three crucial points in the Abstract and in the Introduction of the paper:

  1. How the existence of their findings in mortality data could help practitioners to improve the understanding of mortality data features.
  2. The main contributions of the paper to obtain life tables comparing to previous papers. Authors should describe poor-data where life expectancy at birth is known but not death rates.
  3. If the life tables obtained are accurate enough to evaluate pension systems sustainability.

It would be interesting to know how to implement the model in MortalityLaws R-package; now, it is not in vignettes, maybe authors can add an Appendix. On the other hand, the Reproducible research section is better to include URL https://github.com/mpascariu/MortalityEstimate instead of the general URL https://github.com.

Finally, the author(s) should discuss their approach's real advantage compared to the existing ones. Results should be revised with more detail, and some conclusions or practical interpretations about them should be indicated in the Conclusions. For example, Do the results depend on the country? Are there different results for different age ranges? How does this result help actuaries in pricing? Can models be used in a portfolio?

Particular comments:

Pag 2, lines 57-65. The author(s) should discuss the real advantage of their approach compared to each relational model. Where is the research gap?

Pag 2, lines 74-75. The suggestion is to show poor data countries with life known expectancy at birth that need to predict death rates. Therefore, an algorithm that derives a life table based only on life expectancy at birth can also be widely used in forecasting practice.

Pag 2, lines 16-18. Which are the main conclusions of these papers on relational models? Which countries are studied? In my opinion, the authors enumerate different relational models, which is not enough to justify another paper.

Pag 2, line 73. Please, quote the web and reference of the HMD database.

Pag 2, lines 88-89. Authors should justify the countries' selection based on actuaries or demographers' interests, countries with different demographic and economic evolution, and especially the data-poor context. Results can change with these choices.

Pag 2, line 91 only mortality data between age 0 and 100 are used in the model, but Figure 1 shows age 110. In this figure, sparse data corresponds to age 10 or 20, not only to ages higher to 100. It is worth mentioning that just a graph is subjective to justify linearity; authors should use the Pearson correlation coefficient.

Pag 3, line 101 εxt is not a parameter. Authors 

Pag 3, line 112 Authors should not talk about strong linear trends based on Figures 1 and 2 that depends on graph scale.

Pag 4. May the authors provide parameter significance?

Pag 4 To gain precision in the fitting authors should use confidence intervals for age-specific death rates.

Pag 4. Figure 2 label of y-axis is not life expectancy at birth; it is life expectancy at age 65. Why have not authors done the model excluding young ages? Maybe it more interesting for actuaries.

Pag 5, line 142, which Appendix A or B?

Pag 5, line 151 see Appendix B. On the other hand, minor discrepancies in some examples do not justify the use of least squares over maximum likelihood.

Pag 7, Please, the expression of relative error should be given; results seem very high, especially in Figure 6.

Pag 14-16. Please, revise and update references.

Typing errors

Pag 1: There is a mistake in the reference ? which means a lack of this reference.

Author Response

See file attached.

Round 2

Reviewer 3 Report

The manuscript was improved, and the Authors followed most of my suggestions.
Although, in my opinion, authors should make two changes:
-study mortality in poor-data countries examples
- describe situations or countries where life expectancy at birth is known but not death rates.

Author Response

Please find the author's response for the reviewer enclosed.

Round 3

Reviewer 3 Report

The manuscript was improved, and the Authors describe situations or countries where life expectancy at birth is known but not death rates (in the cover letter), but they did not use those countries in the paper. In my opinion, authors should add this description to the paper.
On the other hand, the study mortality is not in poor-data countries examples- although they recognized limitations, which could be enough.

Author Response

See file attached.
